# Synergistic Field Crop Pest Management Properties of Plant-Derived Essential Oils in Combination with Synthetic Pesticides and Bioactive Molecules: A Review

**DOI:** 10.3390/foods10092016

**Published:** 2021-08-27

**Authors:** Mackingsley Kushan Dassanayake, Chien Hwa Chong, Teng-Jin Khoo, Adam Figiel, Antoni Szumny, Chee Ming Choo

**Affiliations:** 1School of Pharmacy, Faculty of Science and Engineering, University of Nottingham Malaysia, Jalan Broga, Semenyih 43500, Malaysia; hyxmd1@nottingham.edu.my (M.K.D.); tengjin.khoo@nottingham.edu.my (T.-J.K.); 2Department of Chemical and Environmental Engineering, Faculty of Science and Engineering, University of Nottingham, Jalan Broga, Semenyih 43500, Malaysia; 3Institute of Agricultural Engineering, Wrocław University of Environmental and Life Sciences, Chełmońskiego 37a, 51-630 Wrocław, Poland; adam.figiel@upwr.edu.pl; 4Department of Chemistry, Wrocław University of Environmental and Life Sciences, Norwida 25, 50-375 Wrocław, Poland; antoni.szumny@upwr.edu.pl; 5Centre for Water Research, Faculty of Engineering, Built Environment and Information Technology, SEGi University Kota Damansara, Petaling Jaya 47810, Malaysia; choocheeming@segi.edu.my

**Keywords:** phytochemicals, synergism, essential oils, fractional inhibitory concentration, insect mortality rate, phytopathogenic fungi, insect pests, pesticide resistance, fungicide resistance

## Abstract

The management of insect pests and fungal diseases that cause damage to crops has become challenging due to the rise of pesticide and fungicide resistance. The recent developments in studies related to plant-derived essential oil products has led to the discovery of a range of phytochemicals with the potential to combat pesticide and fungicide resistance. This review paper summarizes and interprets the findings of experimental work based on plant-based essential oils in combination with existing pesticidal and fungicidal agents and novel bioactive natural and synthetic molecules against the insect pests and fungi responsible for the damage of crops. The insect mortality rate and fractional inhibitory concentration were used to evaluate the insecticidal and fungicidal activities of essential oil synergists against crop-associated pests. A number of studies have revealed that plant-derived essential oils are capable of enhancing the insect mortality rate and reducing the minimum inhibitory concentration of commercially available pesticides, fungicides and other bioactive molecules. Considering these facts, plant-derived essential oils represent a valuable and novel source of bioactive compounds with potent synergism to modulate crop-associated insect pests and phytopathogenic fungi.

## 1. Introduction

The demand for the production of crops is rising due to the increasing global population, which may exceed 35% by 2050 [1]. This has led to a 15–20-fold use of pesticides in order to enhance the availability of crop yields across the globe [2]. Pesticides are chemical agents that are either synthetically made or naturally occurring, which can be classified as insecticides, fungicides, herbicides, nematicides, rodenticides, etc. Approximately, 2 million metric tons of pesticides are used in agriculture across the globe annually, where countries like China, the USA and Argentina are the major contributors towards pesticide use, and it has been estimated that annual pesticide usage will soon increase up to 3.5 million metric tons worldwide [3]. It has been reported that around 47.5% of herbicides, 29.5% of insecticides, 17.5% of fungicides and the remaining 5.5% of other pest management agents account for all pesticides used worldwide [4]. However, the overuse of synthetic pesticides has led to serious health and ecological hazards, such as the increased risk of cancers, as well as cardiovascular, neurological, endocrine-related health issues and the potential damage done to non-target animals and plants that exist within the parameters of the agent applied [5]. For example, workers who were handling pesticides that consist of hexachlorocyclohexane (HCH) have experienced neurological symptoms. It was reported in 1992 by the National Institute of Occupational Health (NIOH) that paddy field workmen who were spraying insecticides containing methomyl showed abnormalities in their ECG, serum LDH and cholinesterase levels [6]. Chlorpyrifos is one of the most widely used synthetic pesticides in the history of agricultural practices, and the application of this agent can contaminate the soil and groundwater and known to be highly toxic to aquatic life [7]. The environmental, ecotoxicological and health consequences of the widespread application of synthetically made chemical pesticides and fungicides, as well as the development of resistance to these agents, have resulted in a heightened concern and interest among researchers and consumers to focus more on natural and sustainable products with fewer synthetic pesticides, insecticides, fungicides and herbicides [5].The quality of nutrition, food security and sustainability have become very important agenda issues in Sustainable Development Goal 2 (SDG2) established by the United Nations in 2015, and according to current estimates of SDG2, about 8.9% or 690 million people of the world population are in hunger; thus, it may not be possible to achieve zero hunger by the year 2030 [8]. Hence, the United Nations World Food Program aims to alleviate worldwide starvation by the year 2050. There exists a potential to integrate essential oils (EOs) and bioactive compounds from plants, herbs, fruit waste and enzymes of ripening fruits into agricultural practices. Essential oils (EOs) and bioactive compounds from plants, herbs, fruit waste and enzymes of fruits or biomaterials are potential crop protection agents [9]. Essential oils are odoriferous volatile natural oils that can be characterized by their aromatic and lipophilic nature [10]. These EOs are promising sources of naturally occurring bioactive compounds that show pesticidal and fungicidal activities [11]. Plants produce both primary (e.g., sugars and acids) and secondary metabolites, where EOs are largely composed of bioactive secondary metabolites like monoterpenes, esters, sesquiterpenes, phenols, aldehydes, oxides and ketones that are synthesized both internally and externally by plants [12,13]. Essential oils are abundantly found in aromatic plants, where more than 3000 types of EOs have been identified and about 300 essential oil variants have been commercialized [10,11,14,15]. Families of plants that are frequently studied for their essential oils include *Lauraceae*, *Myrtaceae*, *Lamiaceae*, *Rutaceae*, *Apiaceae*, *Asteraceae*, *Poaceae*, *Cupressaceae*, *Piperaceae* and *Zingiberaceae* [16,17,18]. Nonetheless, the demand for novel pesticidal and fungicidal products from natural sources is increasing, and it has been estimated that around 40%–50% of the crop yields of maize, barley, wheat, rice, potatoes, sugar beets and soybeans harvested worldwide are dissipated each year, largely due to pesticide resistance in crop-consuming insects [2]. The registration process for a new fungicide or pesticide usually requires the registrant (e.g., manufacturer) to analyze and conduct different laboratory-based tests [19]. These tests will define the chemistry of the new fungicide of pesticide, as well as the potential hazards to humans, domestic animals, and the proximal environmental and the impact on non-target organisms. Data that include the identity, chemical and physical properties of the active ingredient present in the product, as well as analytical methods, the proposed label and uses, human and environmental toxicity, safety data sheets, efficacy associated with the intended use, container management, residues resulting from the pesticide product usage and the disposal of product waste, are needed to support the application of a pesticide or fungicide registration during its full life-cycle [19,20]. The generation and verification of such data for a single compound may take many years and can be expensive [21]. Hence, there is a growing interest and continuous demand to discover new insecticidal, fungicidal and herbicidal agents with novel mechanisms of action, accompanied by efforts to ensure safety and reduce production cost.

Currently, research has been implemented on various chemical properties and biological activities like antioxidant, anticancer, antimicrobial, antiviral and pesticidal effects of plant-derived essential oils [22]. The following review paper emphasizes the impact of potent plant-derived essential oils and their bioactive compounds that synergistically integrate with synthetic pesticides and other novel molecules for crop preservation.

## 2. Historical Background and Development of Natural Products in Agriculture

Bioactive compounds present in these natural products can be applied as pesticidal, insecticidal and fungicidal agents [23]. The origins of many synthetic pesticidal, insecticidal and antifungal agents can be traced back from a variety of natural products since the introduction and commercialization of penicillin [24,25,26]. The use of plant-based pesticidal agents has been reported since ancient times, where extracts of poisonous herbs were used to control crop-consuming insect pests about 4000 years ago [27]. Nicotine sulfate, extracted from the leaves of tobacco plants, was applied as a natural insecticide in the seventeenth century, and compounds like pyrethrum derived from chrysanthemums flowers and rotenone extracted from the roots of tropical vegetables were used as natural pesticides in the nineteenth century [28]. The use of naturally occurring substances as fungicidal agents has been reported since the seventeenth century, when sea salt and lime were used to treat wheat in order to prevent the growth of bunt caused by fungi [29]. Another important discovery was made by the French botanist Pierre-Marie-Alexis Millardet, who concluded that copper sulfate, which is a naturally occurring substance, was able to effectively control and reduce downy mildew of certain fruits like grapes [30]. Natural products and their bioactive derivatives constituted about 36% of ingredients present in commercially available pesticides from 1997 to 2010. For example, soil-borne bacteria and *Streptomyces avermitilis* and *Saccharopolyspora spinosa* were used to produce natural pesticides known as avermectin and spinosyn, which can effectively cause the paralysis of insect pests [31]. Avermectin is an award-winning natural pesticidal agent that was isolated from the actinomycete species of bacteria known as *S. avermitilis*. Glufosinate, also known as phosphinothricin, is a naturally occurring broad-spectrum herbicidal agent produced by the bacteria of *Streptomyces* spp. [23]. This bacterial-derived compound was commercialized as an herbicide by the German pharmaceutical company named Bayer under the trade name of Finale [32,33]. The herbicidal action of glufosinate works by inhibiting the enzyme glutamine synthetase, resulting in the buildup of ammonia in the thylakoid lumen of plants and leading to photophosphorylation decoupling. The British pharmaceutical company named Corteva Agriscience commercialized a fungicide known as fenpicoxamid that was derived from antimycin, which is naturally produced by *Streptomyces* spp. bacteria. Fenpicoxamid works by inhibiting cellular respiration in fungi. The annual gross of fenpicoxamid and glufosinate exceeded USD 1 billion after introducing them to the market. Other examples of herbicides include the *Streptomyces* spp. produced tentoxin and the fungal *Alternaria alternate* (Fries)-derived thaxtomin [23]. These herbicidal agents were able to disrupt energy metabolism cellulose biosynthesis. Cornexistin is a fungal metabolite derived from *Paecilomyces variotii,* which acts as a broad-spectrum herbicidal agent against maize via the inactivation of enzymes known as aminotransferases [23,34].

## 3. Sources and Chemical Composition of Plant-Derived Essential Oils

Several species of plants consist of volatile essential oils, in which different plant parts like leaves, barks, peels, flowers, seeds, buds and roots can be diverse sources of various essential oils [35]. Plant-based essential oils are complex mixtures of naturally occurring polar and nonpolar compounds [36]. These essential oils have been classified into four primary groups as terpenes, derivatives of benzene, hydrocarbons and other forms of miscellaneous aromatic compounds [37,38]. Terpenes like monoterpenes and monoterpenoids are the most abundant and major representative molecules that constitute about 90% of EOs [39]. Plant-derived EOs are largely composed of carbon hydrocarbons including the following: acyclic alcohols like geraniol, linalool and citronellol; cyclic alcohols like terpeniol, menthol and isopulegol; bicyclic alcohol compounds like verbenol and borneol; phenols that include carvacrol and thymol; ketones like menthone, carvone and thujone; aldehydes that include citral and citronellal;acids like chrysanthemic acid; and oxides like cineole [35]. Terpenes present in these EOs are further classified into the following groups according to their molecular weight: hemiterpenes (C5), monoterpenes (C10), sesquiterpenes (C15), diterpenes (C20), triterpenes (C30) and tetraterpenes (C40). Aromatic compounds occur less frequently compared to terpenes and are natural derivatives of phenylpropane compounds like cinnamaldehyde, aldehyde, cinnamic alcohol, as well as phenols that include eugenol and chavicol, methoxy derivatives like elemicine, methyl eugenols, anethole, estragole and methylenedioxy compounds like myristicine, apiole and safrole [14]. Although EOs are present in a variety of plants, their extraction and productivity are relatively time consuming and expensive processes, since very small amounts of pure EOs can be harnessed from a large amount of raw plant material [35,40].

## 4. Pesticidal and Fungicidal Action Mechanisms of Plant-Derived Essential Oils

Plant-derived essential oils consist of intrinsic properties that can interfere with biochemical, physiological and metabolic functions of insects and fungi by altering the biological activities of target sites of these organisms [41,42]. Anti-insect pest and antifungal agents from botanical EOs can have either narrow-spectrum or broad-spectrum activity, in which narrow-spectrum agents will only affect a particular species of insects or fungi and broad-spectrum agents are effective against a wide range of fungi or insect pests [43]. Additionally, these botanical agents can be classified as fungistatic, which only slow down the growth and multiplication of fungi but do not actually kill them, or asfungicidals that directly promote the cellular destruction of fungal organisms [44]. In case of anti-insect pest plant-derived agents, these can be classified as insect repellents, which consist of chemical properties that can simply repel insects, or as insecticides, which are lethal to insects and cause mortality upon contact [45].

### 4.1. Mode of Action of Insecticidal Essential Oils

Several molecular studies have revealed the action mechanisms of plant-derived essential oils that show the efficiency of pesticidal and insect repellent activity. The EO metabolite-mediated inhibition of acetylcholinesterase (AChE) and octopamine pathways of insects [46,47,48,49,50,51] (Figure 1) has been well investigated and documented. Among these mechanisms, the inhibition of AChE is one of the most exploited, since AChE is an enzyme that plays a crucial role in neuromuscular and neuronal communication in insects [52,53,54]. AChE inhibition can cause neurotransmitter toxic effect on insect pests by the membrane disruption of the postsynaptic junction that leads to the interference of nerve current [55,56,57]. Octopamine is an important hormone associated with the nervous system of insects [58]. This neurohormone is present as octopamine-1 and octopamine-2 and respectively functions as a neurotransmitter and as a neuromodulator in insects, in which the inhibition of octopamine will cause the impairment of physiological modulation associated with muscle juncture and homeostasis of insect bodily fluids, which can alter their octopamine-mediated nervous system [59,60,61,62,63,64]. Plant-derived EOs are also capable of inhibiting GABA receptors present in insects, which can suspend GABA from binding with GABArs (GABA receptors) in extrasynaptic synaptic membranes [65,66,67] (Figure 1). Furthermore, phytochemical metabolites from plant-derived EOs can inhibit or interfere the activities of enzymes associated with the metabolism of xenobiotics and respiration of insects like CarEs, chitin, cytochrome P450s, ATP-binding cassette transporters and GSTs [68,69,70].

### 4.2. Mode of Action of Fungicidal Essential Oils

Plant-derived essential oils have multiple mechanisms of action to inhibit the growth and activity of fungi. Target sites of these EO metabolites include the biosynthesis of cell wall, ATPases activity, efflux pumps, quorum sensing/biofilm formation and cell membrane structure and integrity in fungi [71,72,73,74,75] (Figure 2). Essential oils that disrupt cell wall biosynthesis work by inhibiting the formation of components like chitin and β-glucans, which are necessary for the synthesis of fungal cell walls [76]. Ergosterol is an essential compound associated with fungal cell membranes and their biosynthetic pathways. The inhibition of ergosterol by EOs will cause structural, metabolic and osmotic instability in fungal cells, leading to compromised multiplication and virulence [77,78,79,80]. Certain EOs can affect the ATPases activity of fungi by interfering with the function enzymes associated with fungal mitochondria. The inhibition of mitochondrial enzymes like malate dehydrogenase, succinate dehydrogenase and lactate dehydrogenase can alter the level of reactive oxygen species and ATP, which leads to the diminishing of mitochondrial content that is essential for fungal metabolic pathways [81]. Efflux pumps are proteinaceous transporters localized in the cell membranes of both prokaryotic and eukaryotic cells. In fungi, these are important structures that mediate nutrient uptake, medium acidification and antifungal resistance. These efflux pumps are target sites of certain metabolites associated with plant-derived essential oils in modifying or reversing antifungal resistance [81,82,83]. Plant-derived EOs are also capable of attenuating quorum-sensing (QS) activity in fungi, in which certain phytochemical metabolites present in these essential oils can inhibit cell-to-cell communicating QS signaling molecules like N-acyl homoserine lactones (AHLs), tyrosol, α-(1,3)-glucan and tryptophol, and fungal pheromones like a-factor and α-factor [84,85,86,87,88,89].

## 5. Synergistic and Hybridized Insect Pest Management Products of Botanical Essential Oils

### 5.1. As Homosynergistic Agents

Plant-derived bioactive metabolites present in EOs are capable of interacting synergistically to increase pesticidal action. A study revealed that essential oil phytochemical compounds thymol and 1,8-cineole (Figure 3) interacted synergistically with pulegone to induce larvicidal activity against *Plutella xylostella* (Linnaeus) (diamondback moth). 1,8-cineole and pulegone (Figure 3) combination indicated the highest synergistic activity with a larval mortality rate of 90% in the study. The investigation further elucidated that thymol and 1,8-cineole were able to affect the levels of enzymes like carboxylesterase esterase, glutathione transferases and acetylcholinesterase associated with *P. xylostella* [90]. Rosemary essential oil compounds camphor (Figure 3) and 1,8-cineole indicated synergistic insecticidal action against the moth species known as *Trichoplusia ni* (cabbage looper). The study revealed that the mixture of these compounds (103 µg of 1,8-cineole and 150 µg of camphor) indicated a larval mortality rate >80% in both contact and fumigant assays with a penetration rate >40% in 60 min of application [91]. A similar study conducted by Tak and Isman [92] revealed that 1,8-cineole and camphor isolated from the essential oil of *Rosmarinus officinalis* were synergistically active when combined against *Trichoplusiani* (Hübner) larvae. A compound combination ratio of 60:40 of 1,8-cineole and camphor indicated a larvae mortality rate of 93.3 ± 6.7 in the study [92]. Binary mixtures of essential oil compounds α-terpineol (Figure 3) and thymol were able to synergize the biopesticidal activity of 1,8-cineole and linalool (Figure 3) against swinhoe larvae *Chilopartellus* (Swinhoe) at a dose of 189.7 μg [93]. An investigation conducted by Hummelbrunner and Isman [94] revealed that complex mixtures of *trans*-anethole, citronellal (Figure 3), α-terpineol and thymol were able to interact synergistically and mediate acute toxicity to *S. litura* Fab. (tobacco cutworms) when topically administered at a dose of 40.6 μg [94]. Liu et al. [95] indicated that essential oils extracted from *Cinnamomum camphora* (L.) Presl. seeds and *Artemisia princeps* Pamp leaves exhibited synergistic insecticidal and repellent activity against crop pests like *Sitophilus**oryzae* L. (rice weevil) and *B. rugimanus* Bohem when combined at a concentration ratio of 1:1 [95]. A study showed that cinnamon oil was able to synergize the larvicidal activity of rotenone against *Spodoptera litura* (F.) at a mixture ratio of 1:35 and concentration of 506 mg/L within 72 h of exposure [96]. Essential oil compounds γ-terpinene and terpinen-4-ol (Figure 3) isolated from the extracts of *Majorana hortensis* Moench were able to synergistically mediate insecticidal activity against *Aphis fabae* (Scopoli) and *S. littoralis* [97]. Andrés et al. [98] showed that binary mixtures of essential oil compounds terpinolene and safrole (Figure 3) extracted from *Piper hispidinervum* were able to induce synergistic antifeedant effect on crop-related pests like *Leptinotarsa decemlineata* (Say), *S. littoralis*, *Rhopalosiphum padi*(Linnaeus) and *Myzuspersicae* (Sulzer) [98]. Furthermore, an investigation showed that a binary mixture composed of limonene and carvone (Figure 3) at a concentration ratio of 6:2 displayed synergistic pesticidal activity against *Tribolium castaneum* (Herbst) (red flour beetle) adults at 10.84 µg and larvae at 30.62 µg [99]. Examples of insecticidal homosynergistic plant-derived EOs and their compounds are summarized in Table 1.

### 5.2. As Enhancers of Commercial Insecticides

Certain essential oils and their representative phytochemical constituents are capable of enhancing the insecticidal action of commercially available synthetic chemical pesticides. A study conducted by El-Meniawi et al. [100] showed that EOs from *Simmodsiachinesis*, *Allium sativum*, Fam. and *Mentha piperita* Fam. were able to synergistically enhance the activity of cyhalothrin, diuron and malathion, respectively, at concentrations ranging from 0.1 to 100 µm against *Bemisia tabaci* (Gennadius) (silver leaf whitefly). Further investigations in this study showed that these combinative agents induced the inhibition of the entomic enzymes ATPase, chitinase and acetylcholinesterase [100]. An investigation revealed the pesticide susceptibility of *Myzus persicae* (Sulzer) (green peach aphid) to imidacloprid and spirotetramat after individually combining them with *Thymus vulgaris* and *Lavandula angustifolia*, thymol and linalool, respectively. Imidacloprid with *L. angustifolia* combinative treatment indicated the highest synergism ratio of 19.8 in the study [101]. A similar study showed that rapeseed oil and soya oil enhanced the pesticidal action of pirimicarb and imidacloprid against *Myzuspersicae* (Sulzer) [102]. The essential oil compound linalool isolated from *Ocimumbasilicum* (Linnaeus) enhanced the pesticidal effect of deltamethrin against *Spodoptera frugiperda* (J.E. Smith) (all armyworm). The study showed that the dose of deltamethrin can be reduced by more than 6-fold by the application of 480 μg/μL of *O*. *basilicum* essential oil [103]. Another study indicated that deltamethrin at 9.62 μL and linalool at 0.177 μL combination induced enhanced insecticidal activity against *S. frugiperda* larvae, resulting in 95.75% mortality in 24 hours. The same study showed that linaloolatat0.177 μL enhanced the pesticidal activity of Decis^®^ (25CE) at 0.25 μL, resulting in 100% larval mortality [104]. A recent research study conducted by Ismail (2021) showed that garlic oil was able to synergize and enhance the insecticidal action of chlorpyrifos and cypermethrin up to 9-fold against the crop pest *S. littoralis*. The study further elucidated that these combinative agents induced the inhibition of enzyme pathways associated with oxidase, glutathione S-transferase and general esterase (ά-β-EST) of the tested insect pest [105]. Mantzoukas et al. [106] stated that the cannabidiol (Figure 3) present in the essential oil of the Cannabis plant synergized the commercially available biopesticides madex, azatin and helicovex against the four crop pests *S. zeamais*, *Rhyzopertha dominica* (Fabricius), *Prostephanus truncates* (Horn) and *Trogodermagranarium* (Everts) at doses ranging from 500 to 3000 ppm [106]. Examples of commercially available synthetic pesticides used in combination with plant-derived essential oils and their compounds are summarized in Table 1.

## 6. Synergistic and Hybridized Fungicidal Activity of Botanical Essential Oils

### 6.1. As Homosynergistic Agents

Bioactive phytochemical metabolites present in EOs have been found to interact synergistically to mediate antifungal activity. A study revealed that EOs isolated from thyme, clove and lemongrass demonstrated high antifungal activity, which completely inhibited the growth of mycelium of *Fusarium oxysporum* (Sacc.) and *Fusarium circinatum* (Nirenberg and O’Donnell) at a concentration of 1000 µL/L [107]. Another study indicated that the essential oil combination of thyme, cinnamon, lime and clove induced antifungal activity against the crop-degrading fungus *Colletotrichum gloeosporioides* (Penz)and reduced the damage of crops [108]. Nardoni et al. [109] stated that EOs extracted from *Thymus vulgaris*, *Origanum vulgare*, *O. basilicum*, *Foeniculu mvulgare*, *Illicium verum*, *Syzygium aromaticum*, *Origanum majorana*, *Rosmarinus officinalis*, *Citrus sinensis*, *Citrus bergamia*, *Cymbopogon citrates*, *Salvia sclarea*, *Citrus aurantium*, *Citrus paradise* and *Citrus limon* showed synergistic antifungal activity against *P*. *funiculosum* and *M*. *racemosus* with a FICI of <0.5 for both fungi [109]. An investigation conducted by Bedoya-Serna et al. [110] showed that a nano-emulsion composed of a mixture of oregano and sunflower essential oil was synergistically active against *Fusarium* sp., *Cladosporium* sp. and *Penicillium* sp., which suspended their fungal spore formation at a concentration 0.1 mL [110]. Essential oils extracted from *Thymus vulgaris* and *O. vulgare* interacted synergistically to mediate antifungal activity against *Fusarium* spp. with FICIs ranging from 0.375 to 0.5 when used in combination. Moreover, the study showed that the best synergistic activity of the essential oil combination was demonstrated against *F. moniliforme* with a FICI of 0.375 at an indicative MIC and MFC of 0.156 μL/mL [111]. An investigation carried out by Yen and Chang [112] indicated that cinnamaldehyde and eugenol isolated from cinnamon essential oil were synergistically fungicidal against *L. sulphureus*. The study revealed that the MIC of the cinnamaldehyde and eugenol (Figure 3) combination was90% lower compared to their stand-alone treatments [112]. Hartati [113] stated that combining essential oils extracted from *Cymbopogon nardus* (citronella) and *Azadirachta indica* (neem) at a concentration ratio of 1:1 was synergistic and effective against the fungal pathogen of patchouli plants known as *Synchytriumpogostemonis* S.D.Patil and Mahab [113]. A study revealed that a combination of essential oils from *Syzygium aromaticum* (Linn.) (clove) and *Cinnamonum zeylanicum* (cinnamon) mediated synergistic fungicidal activity against a crop disease causing *Aspergillus niger*, *Alternaria alternate* (Fries) Keissler, *Colletotrichum gloeosporioides* (Penzig), *Lasiodiplodia theobromae* (Patouillard) Griffon and Maublanc, *Plasmopara viticola* (Berkeley and Curtis) and *Rhizopus stolonifer* (Ehrenberg) Vuillemin. The best synergistic antifungal activity was observed for clove oil and cinnamon oil (9:1) with a FICI of 0.55 against *P. viticola* in the study [114]. A research conducted by Yu et al. [115] indicated that essential oil compounds terpinolene, terpinen-4-ol, δ-terpinene, α-pinene, 1,8-cineole, α-terpineol and α-terpinene (Figure 3) isolated from *Melaleuca alternifolia* (tea tree) interacted synergistically to mediate antifungal activity against *Botrytis cinerea* (Persoon). According to the results of the study, the highest antifungal synergism was observed for terpinen-4-ol and α-terpineol combination (1:1 ratio), which indicated a mycelial growth inhibition rate of 99.46% ± 0.76%, and scanning electron microscopic analysis revealed that these compounds made pronounced alterations in the cell wall ultrastructure, mycelial morphology and plasma membrane permeability [115]. Another investigation revealed that the essential oil compounds carvone, apiol and limonene (Figure 3) isolated from the seeds of the *Anathallis graveolens* (Pabst) F. Barros plant were synergistically active against *Aspergillus flavus,* which reduced ATPase and dehydrogenase synthesis, leading to fungal mitochondrial dysfunction and cell death induced by the accumulation ROS in *A. flavus* [116]. Moreover, Nakahara et al. [117] tested the combined activity of the EO compounds linalool and citronellal isolated from *C. nardus* against *Aspergillus* sp., *Eurotium* sp. and *Penicillium* sp., and found the combination to be synergistically fungicidal at a concentration of 112 mg/L [117]. Examples of fungicidal homosynergistic plant-derived EOs and their compounds are summarized in Table 1.

### 6.2. As Enhancers of Commercial Antifungal Agents

Plant-derived metabolites present in EOs are also capable of enhancing the antifungal action of existing synthetic chemical fungicidal agents. A study showed that the EO compound cinnamaldehyde potentiated the fungicidal action of fluconazole against *Aspergillus fumigatus* MTCC 2550 by reducing the MIC of the antifungal agent by up to 8-fold [118]. Gadban et al. [119] demonstrated that essential oil extracted from *Tagetesfilifolia* Lag. potentiated the fungicidal activity of difenoconazole, trifloxystrobin, cyproconazole and carbendazim up to 80% when used in combination against the phytopathogenic fungus *Colletotrichum truncatum* (Schweinitz) Andrus and W.D. Moore [119]. An investigation indicated that EO extracted from *Eupatorium adenophorum* leaves that consist of phytochemical compounds like 10Hα-9-oxo-agerophorone, 9-oxo-10, 11-dehydro-agerophorone and 10Hβ-9-oxo-agerophorone (Figure 3) was able to enhance the fungicidal action of mefenoxam and mancozebagainst *Pythium myriotylum* (Drechsler). According to the results of the study, the EO and mancozeb combination indicated the highest synergistic activity with a fungal mycelia growth rate of 100%, and light and transmission electron microscopic analysis revealed that the EO induced hyphae swelling, cell wall disruption, shortening of the cytoplasmic inclusion and degradation of plasma membrane and cytoplasmic organelles [120]. Camiletti et al. [121] tested and concluded the synergistic action of EO extracted from *Tagetes minuta* L., *Laurus nobilis* L. and *T. filifolia* with iprodione against a major crop-associated fungal pathogen known as *Sclerotium cepivorum* (Berkeley) Whetzel (withe rot). In the study, *T. minuta* in combination with iprodione showed the best synergistic activity, which induced 100% growth inhibition of the fungus. Furthermore, the study elucidated that phytochemical compounds anethole, phenylpropanoids, sphatulenol and estragole (Figure 3) were abundantly present in the EOs of the tested plants [121]. An investigation revealed that EO extracted from *Pogestemon patchouli* mediated partial synergism with synthetic antifungal agents like ketoconazole and amphotericin B against *A. niger* and *A. flavus* with a FICI ranging from 0.52 to 1 [122]. Examples of commercially available synthetic fungicidal agents used in combination with plant-derived essential oils and their compounds are summarized in Table 1.

## 7. Novel Developments in Synergistic Insecticidal and Fungicidal Plant-Derived Essential Oils

Recent developments and novel strategies have been implemented to enhance pesticidal and fungicidal actions of plant-based essential oils. A study indicated that the essential oil compound carvacrol (Figure 3) was able to synergistically interact with the crystalline proteins produced by *Bacillus thuringiensis* MPU B9 and MPU B54 strains to mediate larvicidal activity against *Cydia pomonella* (Linnaeus)(codling moth) and *S. exigua* (beet armyworm moth). The best synergistic larvicidal action was observed at a 1:25000 (MPU B54 protein to carvacrol) concentration ratio, which induced a 96.7% (±3.33%) mortality rate [123]. A similar study elucidated that EOs from *A. indica* containing azadirachtin and *Sinapis alba* were synergistically active against crop pests, like *Spodopteraexigua* (Hübner), *C. pomonella* and *Dendrolimus pini* (Linnaeus), when used in combination with bacterial crystalline toxins of *B. thuringiensis* MPU B9 isolate. Hence, the results of the study indicated a 2-fold increase in larvicidal activity of the combined agents [124]. An investigation conducted by Radha et al. [125] stated that essential oils extracted from *Chenopodium ambrosoides* and *Thymus vulgaris* induced synergism with fungal secretions released by *Beauveria bassiana*(Balsamo) Vuillemin to mediate insecticidal and repellent action against *Callosobruchus maculates* (Fabricius) (*Cowpea bruchid*). According to the results of the study, the highest synergistic interaction was observed with *Chenopodium* oil, which induced a 76% mortality rate of *C. maculates* larvae in 168 h after treatment [125]. Yang et al. [126] tested the insecticidal efficiency of polyethylene glycol-coated garlic essential oil against adult *T. castaneum* and found that these nanoparticles are capable of inducing 100% mortality [126]. An investigation demonstrated that essential oil purified from *Pelargonium graveolens* induced 40% mortality of the *Agrotis ipsilon* (Hufnagel) (dark sword-grass) moth when encapsulated and deployed with solid lipid nanoparticles [127]. Research conducted by Pierattini et al. [128] demonstrated that diatomaceous earth molecules worked synergistically to potentiate the insecticidal activity of *O. basilicum* and *Foeniculum vulgare* against *Sitophilus granaries* (Linnaeus). The combinative treatment indicated a synergistic co-toxicity coefficient that ranges from 1.36 to 3.35 for *F. vulgare* and *O. basilicum* [128]. A novel study demonstrated that orange essential oil interacted synergistically with a baculovirus known as the nucleopolyhedrosis virus to induce enhanced larvicidal activity against *S. littoralis* (the cotton leaf wormmoth) [129]. Furthermore, a novel study conducted by Al-alawi. [130] demonstrated that pine essential oil synergistically interacted with secretions of *B. bassiana* BAU016 fungal isolate to induce enhanced larvicidal activity against *Tetranychus urticae* (Koch) (two-spotted spider mite) [130].

An investigation conducted by Nasseri et al. [131] showed that the EO of *Zataria multiflora* mediated synergistic fungicidal action against *Aspergillus ochraceus*, *A. niger*, *A. flavus*, *Alternaria solani*, *Rhizoctonia solani* and *Rhizopus stolonifer* (Ehrenberg) when loaded and used with solid lipid nanoparticles. The study demonstrated that these combinations inhibited 54%–79% of fungal growth [131]. Luque-Alcaraz et al. [132] tested the antifungal efficiency of chitosan and *Schinus molle* (pepper tree) essential oil conjunctive bio-nanocomposites against *Aspergillus parasiticus* and observed a 40%–50% reduction in fungal cell viability [132]. A study indicated that *M. piperita* EO coated with gold nanoparticles induced synergistically enhanced antifungal activity against *A. flavus* [133]. An investigation revealed that *Satureja khuzestanica* (Jamza) essential oil encapsulated with chitosan nanoparticles induced enhanced fungicidal action against *R. stolonifer* [134]. A research study conducted by Kalagatur et al. [135] elucidated that chitosan nanoparticles mediated antifungal activity against the phytopathogenic fungus *Fusarium graminearum* (Schwabe) when incorporated with the EO of *Cymbopogon martini*, which indicated a MIC of 421.7 ± 27.14 and MFC of 618.3 ± 79.35 ppm. Scanning electron microscopic analysis in the study revealed detrimental changes in the fungal macroconidia and further elaborated antifungal action mechanisms like intracellular reactive oxygen species elevation, depletion of ergosterol content and lipid peroxidation. Moreover, the study revealed the abundance of geraniol (Figure 3) in the EO of *C. martini* [135]. Latha and Lal. [136] demonstrated that secretions produced by micro-algae were able to synergize and potentiate the antifungal action of thyme essential oil against the phytopathogenic fungus *Alternariabrassicae,* which causes a serious disease in pre-harvest and post-harvest broccoli crops [136]. A novel study showed that bioactive secretions of *Bacillus subtilis* B26 isolate synergistically enhanced the antifungal action of EOs obtained from myrtlewood, Leyland cypress needles, orange and lime when used in combination against phytopathogenicfungi *Ophiostoma perfectum*, *Trichoderma* spp. and *A. niger* [137]. Furthermore, a similar study elucidated that the essential oil extracted from *Zingiber officinale var. rubrum* induced enhanced fungicidal activity against an *A. niger* FNCC 6080 isolate when combined with the *Lactococcus lactis* produced bacteriocin lantibiotic known as nisin [138]. Examples of novel bioactive molecules used in combination with plant-derived essential oils and their compounds are summarized in Table 1.

## 8. Concluding Remarks and Future Perspectives

The issue of synthetic pesticide, insecticide and fungicide resistance is expanding rapidly across the globe. Hence, the prospects for the application of existing pesticides and fungicides in the future have become challenging and uncertain. Plant-derived essential oils and their phytoconstituents are remarkable sources of novel bioactive compounds with broad-spectrum insecticidal and antifungal properties. These compounds can exert homosynergistic action or synergistically interact with other pest management agents or bioactive molecules. This review summarizes and interprets the findings of experimental work based on plant-based essential oils in combination with existing pesticidal, insecticidal and fungicidal agents, as well as novel bioactive natural and synthetic molecules, against insect pests and fungi responsible for the spoilage of crops. These essential oil combinations have shown remarkable results as agents with different mechanisms for overcoming pesticidal, insecticidal and fungicidal resistance. For instance, several studies have elucidated that these synergistic combinative compounds can significantly reduce the insect mortality rate and MIC/MFC of fungi. The efforts in synergy research have led to the discovery and production of novel pest management agents. However, the underlying modes of actions associated with synergistic essential oil products have not yet been fully exploited. Hence, the broadening of molecular and biochemical studies based on combined synergists of essential oils are needed to establish a better understanding and further exploitation of their toxicological responses and bioactivity in order to determine their true potency and safety in agricultural application. At present, the availability of experimental data based on essential oil synergists is limited and, therefore, further studies are needed in order to broaden and elucidate their novel action mechanisms in modifying pesticidal, insecticidal and fungicidal resistance. Moreover, studies are needed on insecticidal and fungicidal activities of fruit waste and botanical enzymes, like bromelain combinative synergists, with plant-derived essential oils.

## Figures and Tables

**Figure 1 foods-10-02016-f001:**
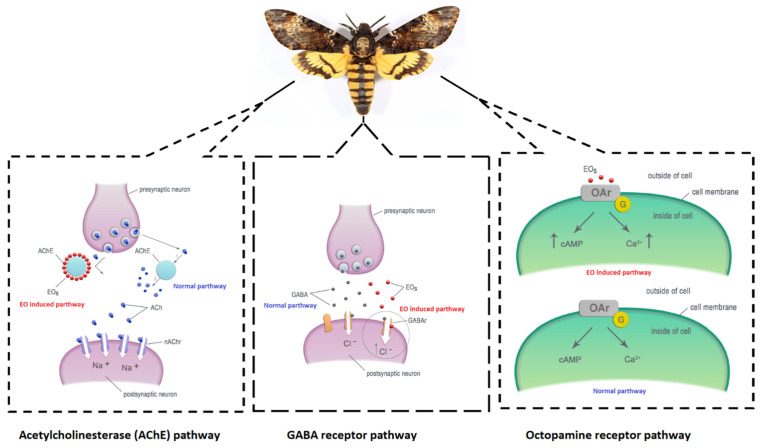
Insecticidal action mechanisms of plant-derived essential oils (e.g., [51]).

**Figure 2 foods-10-02016-f002:**
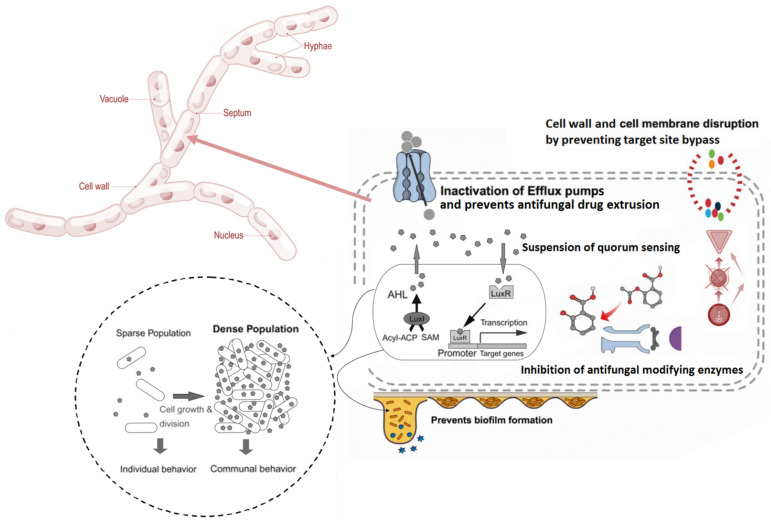
Antifungal action mechanisms of plant-derived essential oils.

**Figure 3 foods-10-02016-f003:**
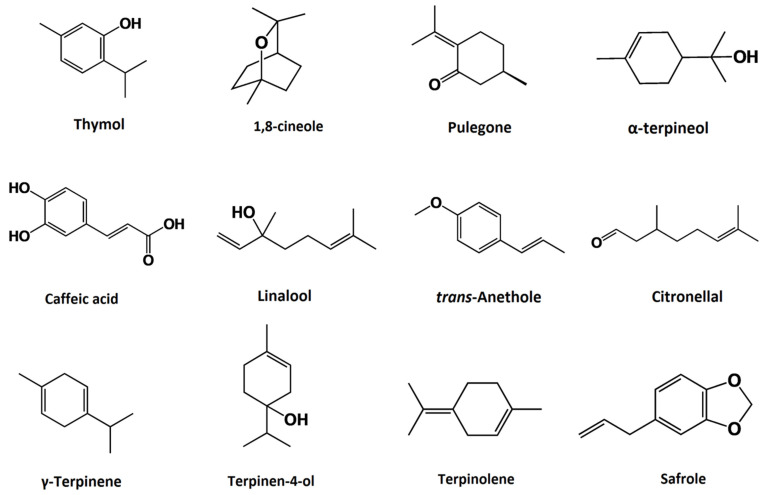
Phytochemical compounds isolated from plant-derived essential oil synergists.

**Table 1 foods-10-02016-t001:** Plant-derived essential oils and their compounds combined with synergistic agents against paddy field insect pests and fungal pathogens.

Plant Source	EO Compound	Synergist Used with EO Compound	Insect Pest//Fungal Pathogen	Reference
N/S	Cinnamaldehyde	Eugenol	*L. sulphureus*	[112]
N/S	Cinnamaldehyde	Eugenol	*C. nardus*	[113]
*Melaleuca alternifolia*	Terpinolene, Terpinen-4-ol, δ-Terpinene,	α-pinene, 1,8-cineole, α-terpineol	*B. cinerea*	[115]
*Anathallis graveolens*	Carvone, Apiol	Limonene	*A. flavus*	[116]
*Cymbopogon nardus*	Linalool	Citronellal	*Aspergillus* sp., *Eurotium* sp., *Penicillium* sp.	[117]
*Tagetes minuta*, *Laurus nobilis*, *Tagetes filifolia*	Anethole, Phenylpropanoids, Sphatulenol, Estragole	Iprodione	*S. cepivorum*	[121]
N/S	Carvacrol	Crystalline proteins of *B. thuringiensis*	*C. pomonella, S. exigua*	[123]
*Azadirachta indica* *Rosmarinus officinalis*	Azadirachtin Camphor	Crystalline toxins of *B. thuringiensis* 1,8-cineole	*S. exigua*, *C. pomonella*, *D. pini* *T. ni*	[124] [91]
N/S	α-terpineol	Thymol	*C. partellus*	[93]
N/S	*Trans*-anethole, Citronellal	α-terpineol, and thymol	*S. litura*	[94]
N/S	Cinnamon oil	Rotenone	*S. litura*	[96]
N/S	Terpinolene	Safrole	*L. decemlineata*, *S. littoralis*, *R. padi*, *M. persicae*	[98]
*Piper hispidinervum*	γ--terpinene	Terpinen-4-ol	*A. fabae*	[97]
			*S. littoralis*	
*Majorana hortensis*	Cinnamaldehyde	Fluconazole	*A. fumigatus*	[118]
N/S N/S *Simmodsia chinesis*	10Hα-9-oxo-agerophorone, 9-oxo-10, 11-dehydro-agerophorone, 10Hβ-9-oxo-agerophorone Jojoba oil	Mefenoxam, Mancozeb Cyhalothrin	*P. myriotylum* *B. tabaci*	[120] [100]
*Allium sativum*	Garlic oil	Diuron		
*Mentha piperita*	Peppermint oil	Malathion		
*Thymus vulgaris*	N/S	Imidacloprid, Spirotetramat	*M. persicae*	[101]
*Lavandula angustifolia*	N/S	Deltamethrin	*S. frugiperda*	[103]
N/S	Linalool, Thymol	Decis^®^ (25CE)	*S. Littoralis*	[104]
*Ocimum basilicum*	Linalool	Chlorpyrifos, Cypermethrin	*S. zeamais*	[105]
		Madex, Azatin, Helicovex	*R. dominica*	[106]
N/S	Garlic oil		*P. truncatus*	
N/S	Cannabidiol oil		*T. granarium*	

N/S: Not specified.

## Data Availability

The following review was based on data extracted from published research articles available in all relevant databases with no limitation up to 10 June 2021.

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
