# Peer review of "Synergistic Field Crop Pest Management Properties of Plant-Derived Essential Oils in Combination with Synthetic Pesticides and Bioactive Molecules: A Review"

_foods, 2021, doi:10.3390/foods10092016_

Round 1

Reviewer 1 Report

The presented manuscript report a comprehensive review on the use of natural products and essential oils as pest management and their combination with synthetic pesticides to work synergistically and induce an enhancement of the synthetic pesticide activity.

The work represents an interesting addition and it will be valuable to highlight a growing interested in essential oils and their application in crop protection. 

However, there are critical elements that need to be fixed.

The authors need to provide proper literature references. The introduction and other part of the manuscript lack of proper references and important citations. In more than one case, authors forgot to follow important affirmation/statement with related references.

Also, it is not clear if authors allocated DDT among naturally derived pesticides. DDT was in fact synthesized from scratch in 1874 by an Austrian chemist.

There are lots of typos and authors need to check carefully the format of scientific names. Many species are not reported with their full name when cited for the first time in the text. Or the same species is repeatedly reported with the corresponding full scientific name.

More specific comments are included in the attached pdf.

Author Response

Dear reviewer. kindly refer to the attached response report.

Reviewer 2 Report

Please read my comments provided. Introduction section and section 2 page 3 Historical background and development of natural products in agriculture. And Section 3 Sources and chemical composition of plant -derived essential oils, need thoroughly edited for citations reasons. In the manuscript  some insect scientific names has authorities inserted ,however other pages where scientific names mentioned no authorities  inserted. This should be consistent. 

Author Response

Dear reviewer. Kindly refer to the attached response report.

Reviewer 3 Report

The manuscript is very interesting. It is well written and  easy reading. 

please do minor changes:

line 22: morality?????Insect morality rate 

Please identify the phytochemical compounds isolated from plant-derived essential oil with the specific plant-derived... . a figure with the compounds and not correlation with the plants is not enough.

Author Response

(The authors gave the same response as above.)

Round 2

Reviewer 2 Report

I would like to congratulate the authors carrying out all correction using different colours and providing a written statement related to the corrections that made my checking easy. However, they missed one error in the Introduction section in first paragraph line 42 please see attached manuscript. 

"Counties" should be "countries" 

Author Response

Dear reviewer. Thank you very much for detecting this error. We have corrected it as suggested.